# Lean4Physics: Comprehensive Reasoning Framework for College-level Physics in Lean4

**Yuxin Li**[1]*, **Minghao Liu**[1]*, **Ruida Wang**[2]*, **Wenzhao Ji**[1],
**Zhitao He**[1], **Rui Pan**[2], **Junming Huang**[3], **Tong Zhang**[2], **Yi R. (May) Fung**[1,4]†
[1]Hong Kong University of Science and Technology,
[2]University of Illinois Urbana-Champaign,
[3]Princeton University
[4]MMSense Lab
{ylinq,mliuby,wjiab,zhebu}@connect.ust.hk
{ruidaw,ruip4}@illinois.edu
junminghuang@princeton.edu
tongzhang@tongzhang-ml.org
yrfung@ust.hk

## Abstract

We present Lean4PHYS, a comprehensive reasoning framework for college-level physics problems in Lean4. To establish a solid foundation for formal reasoning in physics, Lean4PHYS launches *PhysLib*, a repository containing fundamental unit systems and essential theorems to formulate physics proofs in Lean4. It will be community-driven and long-term maintained. Lean4PHYS also includes *LeanPhysBench*, a college-level benchmark for evaluating LLMs' Lean4 formal physics reasoning capability. It contains 200 hand-crafted and peer-reviewed Lean4 theorem statements formalized from university textbooks and physics competition problems. Based on the *PhysLib* and *LeanPhysBench* we composed in Lean4PHYS, we perform exhaustive experiments of baseline results using major expert Math provers and state-of-the-art closed-source models, and provide an analysis of their performance. In the experiment, we identify that most expert provers do not outperform general models as they did in the math domain. This suggests potential overfitting to the math domain rather than learning formal reasoning for formal provers. We also conduct a comprehensive experiment showing that, with *PhysLib* in the context, LLMs' performance on *LeanPhysBench* increases by **11.90%** on average, proving the effectiveness of our repository in assisting LLMs in solving the Lean4 physics problem. To the best of our knowledge, we are the first study to provide a physics benchmark in Lean4. The code for this project will soon be released in https://github.com/ShirleyLIYuxin/Lean4PHYS.

## 1 Introduction

Formal thinking capability has always been considered a cornerstone of human intelligence and a key objective of machine learning. With the emergence of Large Language Models (LLMs), many studies explore diverse ways to apply LLMs to perform various reasoning tasks. This includes general reasoning (Wang et al., 2024b; Suzgun et al., 2022; Talmor et al., 2018), math reasoning (Hendrycks et al., 2021; Cobbe et al., 2021; Guo et al., 2025), natural science reasoning (Saikh et al., 2022; Edwards et al., 2025), and many other domains (He et al., 2025a; Su et al., 2025). However, most works handle reasoning as a pure Natural Language (NL) task that relies on the answer checking to judge the correctness of reasoning, while being unable to verify the intermediate reasoning steps.

---

*First Authors

†Corresponding Author

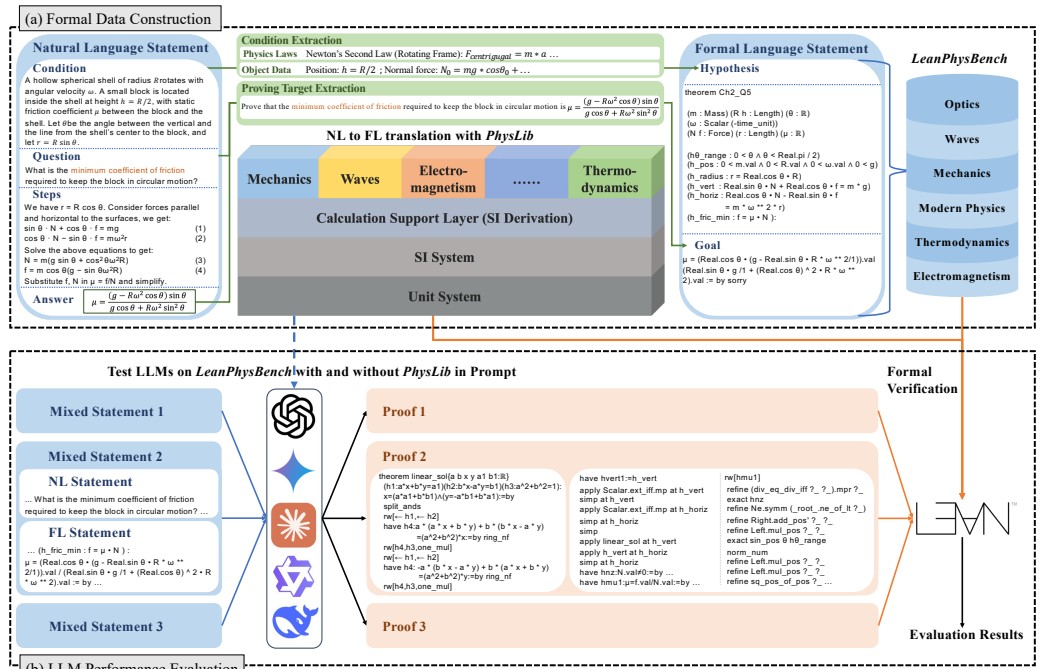

Figure 1: **Overview of the Lean4PHYS framework:** (a) Benchmark and Library Construction: The benchmark and library are developed using a bottom-up principle. We first establish the foundational units and SI system of *PhysLib*, then compose the calculation support, and finally construct the field-related theorems. For the benchmark, NL questions are transformed from a question-answering format to proof problems and subsequently formalized into Lean4 statements. (b) LLM Performance Evaluation: We evaluate the formal physics reasoning capabilities of major open and closed-source LLMs on *LeanPhysBench*, both with and without *PhysLib*. After inference is completed, Lean4 auto-verification evaluates model performance and highlights the utility of *PhysLib*.

To make the reasoning process verifiable, researchers attempted to ground the reasoning procedure in formal logical systems, enabling the automatic verification of both the reasoning process and its results through Formal Languages (FLs). Through this idea, many FLs are developed, including Lean (De Moura et al., 2015; Moura & Ullrich, 2021), Coq (Coq, 1996), Isabelle (Paulson, 1994), and HOL (Harrison, 2009). Among them, Lean4 has received superior attention from both the academic and industry in recent days, making it one of the most well-studied formal languages in recent years. There are numerous benchmarks (Zheng et al., 2021; Gulati et al., 2024; Azerbayev et al., 2023), dataset (Dong & Ma, 2025; Wang et al., 2024a; Ying et al., 2024), and provers (Polu et al., 2022; Wang et al., 2024a; Xin et al., 2024; Wang et al., 2025c; Lin et al., 2025b; Dong & Ma, 2025; Xin et al., 2025; Ren et al., 2025) have been proposed.

However, current studies in formal reasoning primarily focus on the mathematical domain, leaving other domains that can also be formalized, such as physics, largely understudied. Moreover, most of the state-of-the-art expert provers claim their capability in formal reasoning by showing superior results in Lean4 math benchmarks. This raises concerns about whether such formal capability can be transferred to other domains, like physics problems represented using similar Lean syntax, or current provers overfit to mathematical reasoning.

To answer the problems above, we propose Lean4PHYS, a comprehensive reasoning framework for college-level general-domain physics problems in Lean4. Lean4PHYS aims to provide a foundation for LLM-based formal physics reasoning in Lean4. The overview of Lean4PHYS is presented in Figure 1 The framework launches *PhysLib*, a foundation repository that supports formal physics reasoning. It includes a systematic treatment of physical units through a dedicated *UnitsSystem*, enabling dimensionally consistent symbolic computation, along with a growing collection of extendable formalized physics theorems. Based on the *PhysLib*, we develop *LeanPhysBench*, a college-level benchmark to evaluate LLMs' performance in Lean4 physics reasoning. The bench-

mark contains 200 manually-crafted Lean4 formal theorems whose difficulty level varies from standard university exercises to Olympiad-style competition problems. We construct the benchmark by systematically transforming natural-language problems into Lean4 theorems: extracting key conditions and relevant physical laws, defining explicit proving targets, restating the problems in logical form, and integrating them into verifiable Lean4 statements. To the best of our knowledge, *LeanPhysBench* is the first benchmark that evaluates LLMs' formal physics reasoning capability.

We summarize our contributions as follows:

1. We introduce *Lean4PHYS*, a comprehensive Lean4-based framework for formal physics reasoning. It provides *PhysLib*, a modular physics library supporting unit-aware calculations and extensible physical theorems.

2. Our framework innovates by bridging natural-language physics problems with formal Lean4 representations, enabling LLMs to learn domain-specific laws and reasoning patterns beyond standard math-oriented theorem provers.

3. We use *LeanPhysBench* to evaluate the leading model's performance in formal physics reasoning. The experiments suggest that all current expert math provers and general models, regardless of their size, achieve suboptimal performance. Furthermore, we demonstrate that after integrating *PhysLib*, the models exhibit consistent performance enhancements. The results also indicate the potential for overfitting to the math dataset for the current expert Lean provers. When testing only on the hard dataset, we found that all models perform poorly, indicating the current limitations for formal physics, especially statements involving complex calculation-based mathematical operations such as integrals and derivatives.

Moreover, to the best of our knowledge, Lean4PHYS is the first work that tries to extend the LLM-based formal reasoning from math to a more general domain, which offers a new direction that attempts to formalize progressively more subjects.

## 2 THE LEAN4PHYS FRAMEWORK

In this section, we introduce the design and implementation of the Lean4PHYS framework in detail. The core idea of our framework is to provide a foundation and then evaluate the LLMs' formal physics reasoning capabilities. We firstly initiate *PhysLib*, a community collaborative foundation repository for Lean4 physics reasoning in Section 2.1. Subsequently, we present the details of how we construct *LeanPhysBench*, (to the best of our knowledge) the first benchmark for formal physics reasoning.

### 2.1 PHYSLIB

In this paper, we introduce *PhysLib*, a community-driven repository designed to support rigorous and machine-verifiable Lean4 formal physics reasoning. We build the library in a bottom-up manner for both the conceptual level of knowledge and the technical level of implementation. The current version of *PhysLib* contains the foundation of physics unit systems (Section 2.1.1), practical theorems to use in proving (Section 2.1.2), and a guideline for community development and extension (Section 2.1.3).

#### 2.1.1 FOUNDATION OF PHYSICS: UNIT SYSTEM

The central challenge of formalizing physics from scratch is to identify the unchanged kernel that supports all the reasoning in the field. Unlike mathematics, where every basic building block is purely based on definition, physics, as well as other natural sciences, are primarily supported by empirical rules induced from experiments, which are naturally not as clearly defined as mathematics. For physics, we identify such a kernel as the unit system. Thus, we lay the foundation for physics reasoning by establishing the unit system.

In the implementation, we build the unit system by extending Mathlib (mathlib Community, 2020) with the International System of Units (SI) following Tao (2025). The system contains seven basic units, including time, length, mass, electric current, thermodynamic temperature, amount of substance, and unit of luminous intensity (Newell et al., 2019). Based on the unit system foundation,

we introduce the first-order derivative over the most general and commonly used sections, namely, length, taking the derivative over time, introducing velocity.

### 2.1.2 TOPIC-BASED THEOREM DEVELOPMENT

Building upon the cross-topic kernel foundation system of units, we introduce topic-based theorem systems according to different needs in specific kinds of problems. Specifically, the current version of *PhysLib* splits problems into six major topics, namely: mechanics, waves & acoustics, thermodynamics, electromagnetism, optics, and modern physics. The topic split is inspired by Young & Freedman (2019). Our implementation principle for this section of *PhysLib* is to first create different namespaces and independent Lean files for each topic. Subsequently, we add topic-specific unit types and constants in the topic namespace. Then, we implement basic physics rules summarized from experiments as definitions. Finally, we implement theorems with their proofs that are practical to the topic as the final layer. We implement the mechanics field in detail as an example and set a basic foundation for other topics. We present the design process to the community and launch this project for collaborative development of the field.

### 2.1.3 COMMUNITY-DRIVEN AND EXTENSIBILITY

As mentioned before, *PhysLib* is designed to be a community-driven and collaborative work like Mathlib (mathlib Community, 2020), and we make our best effort to ensure other researchers can easily read and extend the system while maintaining consistency. In general, we organize *PhysLib* in a three-layer hierarchy: (1) Foundation unit system, which should be consistent with changes only necessary. (2) Topic-specific unit system, which should be added when formalizing theorem statements where current units are unable to support the construction. (3) Topic-specific theorems, which include most of the practical theorems to support proof implementation and should be regularly updated.

The current version contains a relatively comprehensive implementation of the repository in mechanics, serving as an example for the community. *PhysLib* inherits components from Tao (2025), which are distributed under the Apache-2.0 license, and the same license applies to newly added content. We will strongly encourage contributions of new statements and proofs, and will continuously maintain the repository in the future.

Beyond the current repository we are implementing, such a layered design is applicable to the formalization of other domains, such as the natural and social sciences, and be alternatively extended to general proving systems based on the same logic.

## 2.2 LEANPHYSBENCH

With the current trend of utilizing LLMs to perform formal reasoning, it is crucial to evaluate their formal physics reasoning capability. However, to the best of our knowledge, there is no dataset for benchmarking the Lean4 physics reasoning capability for LLMs. Thus, we propose (to the best of our knowledge) the first benchmark for Lean4 physics reasoning. In this section, we detail the process of creating this benchmark. We firstly present the data collection process in Section 2.2.1, then detail how we created the benchmark in Section 2.2.2, and report the benchmark statistics in Section 2.2.3.

### 2.2.1 DATA COLLECTION & PREPROCESSING

Corresponding to the bottom-up principle, when we design the *PhysLib*, when creating the dataset, we follow a basic-to-advanced principle. Our benchmark primarily consists of two levels of data: the high school Olympiad-competition-related level and the college level. The Olympiad data are collected from competition-related exercise books. This level of data focuses on testing the model's capability to perform multi-step reasoning within a specific field of knowledge. On the other hand, the college-level problems are collected from the elementary university physics textbooks. These problems are selected to cover a deeper range of concepts with easier reasoning steps that focus on testing the LLM's reasoning when the topics span multiple physics models. For questions accompanied by figures, we extract the information from the figures and describe it in natural language, alongside the problems.

Figure 2: Two examples from *LeanPhysBench* demonstrating different difficulty levels.

Following *PhysLib*'s design, we further divided the topic of the *LeanPhysBench* into mechanics, waves & acoustics, thermodynamics, electromagnetism, optics, and modern physics. After the data collection, we have the base natural language statement for formulating *LeanPhysBench*.

### 2.2.2 Formalization pipeline

Following the collection and preprocessing of the NL data, we apply a strict pipeline to transform the NL data into verifiable Lean4 theorems. The overview of the formalization pipeline is shown in Figure 1(a). Formally, if we denote a physics problem we want to formalize by $P$, the original problem we have is $P_{original}$, the target is to obtain the Lean4 version of the problem $P_{Lean}$. The formalization process is as follows:

**NL Format Alignment**  According to previous work in formalizing mathematical statements (Wang et al., 2025b; Zheng et al., 2021), there is a significant representation gap between Lean4 statements and their corresponding NL problems in the original datasets. Specifically, in NL problems, it is typical for the problems to be in question-answering format, where the target is to find a specific numerical or formulaic answer. However, in Lean4, the problem type is a closed-end proof rather than a specific answer, which leads to a gap in the statement. Inspired by Wang et al. (2025b), we perform a format alignment to transform the question-answering style physics problem into a proof statement.

The first step in format alignment is to restate the question-answering problem in a proof format. Specifically, we transform the question part of "Finding the answer to ..." into "Prove that the answer is ..." following Zheng et al. (2021). Subsequently, to better model the physical process and relations, we require the LLM to write a step-by-step solution to the question. Based on the solution, we extract the physics laws used in the problems and all the initial conditions for composing statements. Finally, we define the proving target for the problem. We split the target of proof into three categories, namely, numerical value, physical expression, or logical formula describing a physical state and use such targets to assist the process of Lean4 code writing. After this step, we align the NL question-answering problem to a provable question with extracted physics laws, initial conditions, and proving target.

**Lean4 code writing & Verification**  After we obtain aligned NL problems, the Lean code writing is split into the formalization of conditions and goals. During the process of writing Lean4 conditions for the problem, we add the necessary physics laws to *PhysLib* as well as properly present such laws in Lean code. In formalizing goals, we write the corresponding Lean4 expression of the proving targets as goals for the Lean4 proof. After we obtain the Lean4 code, it is submitted to the verifier to verify whether it compiles successfully. Additionally, we ask experts for physics and Lean to check the completeness of the theorem. Each statement requires one expert to formalize and at least two experts to verify.

The above pipeline ensures that problems in *LeanPhysBench* accurately capture the underlying physics semantics from NL descriptions. We present two examples of Lean4 statements we formalized in Figure 2. In the example, we can clearly see that the college-level problems focus on relatively simple operations in a broader range of topics. Competition-level statements emphasize derivations of formulas using more advanced math tools in a more concentrated field of physics.

### 2.2.3 BENCHMARK STATISTICS

The detailed statistics for the *LeanPhysBench* are presented in Figure 3. In total, *LeanPhysBench* contains 200 physics statements formalized in Lean4. Among them, 104 statements are at the college-level and 96 are at the competition level. The competition level is further divided into easy (62 problems) and hard (34 problems) categories. Where easy problems focus more on mathematical deduction with looser conditions, and hard problems focus more on physics problem deductions with more physics formulas and tighter conditions.

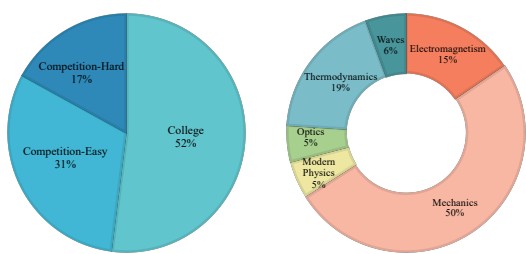

Figure 3: Statistics of *LeanPhysBench*: The distribution of 200 Lean4 physics statements across difficulty levels and topics.

## 3 EXPERIMENT

We conduct comprehensive experiments on Lean4PHYS to demonstrate the significance of our proposed *LeanPhysBench* and the effectiveness of *PhysLib*. In particular, we demonstrate the performance of major expert Lean provers and general models on *LeanPhysBench* and prove the effectiveness of *PhysLib* in Section 3.2. We study the problem format of *LeanPhysBench* in Section 3.3, and in Section 3.4 we perform case studies to address important concerns of the main results.

### 3.1 EXPERIMENT SETUP

We evaluate the LLMs' Lean4 physics reasoning capabilities by applying them to write proofs for the *PhysLib*. Specifically, the task for the LLMs is to compose Lean4 proofs with provided NL statements and Lean4 statements. To ease the load of LLMs in proving, we manually configure all the imports and namespaces. Furthermore, unless otherwise specified, we allow the LLM to use Long Chain-of-Thought capability to perform deeper reasoning. We apply the pass@16 metric to evaluate the performance.

To better demonstrate current LLMs' formal physical reasoning capability, we select the most representative closed-source and open-source models to evaluate. Namely, for closed-source general LLMs, we select GPT-4o (OpenAI et al., 2024), Claude-Sonnet-4 Anthropic (2025), and Gemini-2.5-Pro (Comanici et al., 2025) as baselines. For open-source models, we present both the results of general-purpose models, including DeepSeek-R1-0528 (Guo et al., 2025) and Qwen3-8B (Team, 2025), as well as expert Lean4 provers such as Goedel-Prover-V2-8B (Lin et al., 2025b), Kimina-Prover (Wang et al., 2025a), and DeepSeek-Prover-V2 (Ren et al., 2025).

Furthermore, to demonstrate the effectiveness of *PhysLib*, we test the LLM's capability under modes that have or do not have *PhysLib* in the generated context. In summary, we test the *LeanPhysBench* on eight major LLMs in two modes. We note that the experiments were conducted using the initial version of *PhysLib* and all reported results correspond to the initial version. Both the initial version and the updated version of *PhysLib* will be publicly released. The implementation details of our experiments can be found in Appendix B.

### 3.2 MAIN RESULT

We demonstrate our main experiment result in Table 1. The result demonstrates that LLMs that are larger in size and have better coding capabilities, such as Gemini and Claude, perform better in formal physics reasoning. With *PhysLib* in context, Gemini achieves a comparatively higher accuracy rate of 40.0% for the entire dataset, while Claude obtains 34.50%. In comparison, the expert provers

Table 1: Pass@16 results of *LeanPhysBench* on 8 LLMs with (✓) and without (✗) *PhysLib* mode across different difficulty levels, including College, Competition-Easy (Comp-Easy), and Comp-Hard. All results are averaged over **3 runs**. Best mean is **bolded**, second-best is underlined. Standard deviation is shown unformatted.

| Method | PhysLib | College | Comp-Easy | Comp-Hard | Overall |
|---|---|---|---|---|---|
| *Open-source models* | | | | | |
| DeepSeek-R1-8B | ✓ | 6.09% ± 0.45% | 8.60% ± 0.76% | 0.00% ± 0.00% | 5.83% ± 0.47% |
| | ✗ | 2.56% ± 1.81% | 5.91% ± 0.76% | 0.98% ± 1.39% | 3.33% ± 0.94% |
| Qwen3-8B | ✓ | 6.41% ± 0.91% | 10.22% ± 1.52% | 0.00% ± 0.00% | 6.52% ± 0.81% |
| | ✗ | 2.58% ± 0.47% | 3.23% ± 0.00% | 0.00% ± 0.00% | 2.35% ± 0.25% |
| Kimina-Prover-8B | ✓ | 9.94% ± 0.91% | 23.12% ± 1.52% | **6.86%** ± 1.39% | 13.50% ± 0.71% |
| | ✗ | 5.77% ± 0.00% | 17.20% ± 0.76% | 5.88% ± 0.00% | 9.33% ± 0.24% |
| Goedel-Prover-V2-8B | ✓ | 10.58% ± 1.36% | 26.34% ± 1.52% | 5.88% ± 0.00% | 14.67% ± 1.18% |
| | ✗ | 6.09% ± 0.45% | 19.35% ± 0.00% | 4.90% ± 1.39% | 10.00% ± 0.00% |
| DeepSeek-Prover-V2-7B | ✓ | 8.01% ± 0.45% | 31.18% ± 1.52% | 5.88% ± 0.00% | 14.83% ± 0.24% |
| | ✗ | 6.09% ± 0.45% | 22.58% ± 0.00% | 5.88% ± 0.00% | 11.17% ± 0.24% |
| *Closed-source models* | | | | | |
| GPT-4o | ✓ | 9.29% ± 2.40% | 30.11% ± 2.01% | 0.00% ± 0.00% | 14.17% ± 0.85% |
| | ✗ | 2.24% ± 0.45% | 4.30% ± 2.01% | 0.00% ± 0.00% | 2.50% ± 0.41% |
| Claude-Sonnet-4 | ✓ | 29.49% ± 0.45% | 61.83% ± 1.52% | 0.00% ± 0.00% | 34.50% ± 0.41% |
| | ✗ | 3.21% ± 1.63% | 5.91% ± 1.52% | 0.00% ± 0.00% | 3.50% ± 1.22% |
| Gemini-2.5-pro | ✓ | **32.69%** ± 0.79% | **74.19%** ± 1.32% | 0.00% ± 0.00% | **40.00%** ± 0.71% |
| | ✗ | 6.09% ± 0.45% | 11.29% ± 1.32% | 0.00% ± 0.00% | 6.67% ± 0.71% |

and GPT-4o suffer from suboptimal performance with an overall accuracy rate of around 10%. Furthermore, the results demonstrate that with *PhysLib* added to the context, performance improved by **11.90%** and such improvement is consistent across all models and difficulty levels. It indicates the effectiveness of our *PhysLib* in assisting LLMs' formal physics reasoning.

Upon closer examination, we can observe that Lean experts, which significantly outperform closed-source general LLMs in the math domain, lack the strong formal physics capabilities of these models. This reveals that the expert Lean provers' capabilities are limited to the math domain and fail to transfer to other general domains, especially when these domains apply a new definition (such as the unit system in Lean4PHYS). Moreover, we found that the performance difference between expert provers is relatively marginal, indicating that significant improvement in mathematics does not guarantee a large improvement in physics reasoning.

When analyzing the results from different levels of difficulty, we find that the models generally perform well on easy problems in competition problems, which are more closely aligned with mathematical representations. However, expert models still do not outperform larger closed-source models, indicating the limited transfer of capability from math to physics formal reasoning in Lean4. Such a finding is also true for college-level problems, where the performance of models is generally lower. However, for the competition-hard level of problems, the expert provers perform better than the closed-source models. This is because the expert models have a stronger capability to perform complex deduction, whereas large models do not.

Furthermore, adding *PhysLib* to the context will significantly improve the LLMs' performance. This is because when adding the *PhysLib*, the model has a better understanding of the formulation of the basic system we use to compose the proof. Without the *PhysLib*, the model can only perform basic simplification tactics like: `constructor`, `rw`, `abel`, `exact`, `aesop`. By adding the *PhysLib*, the model can learn to perform more advanced tactics, such as `simp` and `norm_num`, based on the theorems and definitions in *PhysLib*.

## 3.3 PROBLEM FORMAT STUDY

This section presents a more detailed study of the problem format. Due to the space limit, we place the example in Figure 4 in the Appendix and only provide the analysis result here.

From the example of problem format, we observe that college-level statements primarily involve numeric computations with a relatively wide range of physical quantities with units. These problems rarely require multi-step formula derivations, but heavily depend on the unit system in *PhysLib* to ensure dimensional consistency. Thus, the models with weaker in-context learning perform relatively badly on this level of problems. This is because they cannot perform novel syntax adaptation or infer unit-handling rules from context.

The easy-level competition questions are closer to transitional math problems in Lean, such as MiniF2F (Zheng et al., 2021). They involve relatively simple formula derivations, typically within two steps and often only tactics from Mathlib. Therefore, the models that are more familiar with the Lean mathematics and good at using tactics perform better than closed-source large models on these kinds of problems even without *PhysLib*.

On the other hand, the hard level of competition problems demands complex symbolic reasoning, handling quantifiers, and difficult functional reasoning. For instance, this includes proving the existence of a number, holding of an inequality, or deriving a functional relation. These problems combine unit casts with symbolic manipulation, further increasing the difficulty. Solving this level of problem requires careful decomposition of the problem, proving new lemmas, and diverse proof strategies with creativity. Moreover, many problems of this level require calculus concepts such as limits and continuity. All the above factors combined cause the low pass rate at this level.

## 3.4 CASE STUDY

We present the case analysis in this section to provide a more detailed examination of the key findings from our main experiment. Due to the space limit, for a detailed example, please refer to Appendix E.2.

**Behavior of the same theorem with and without *PhysLib*** Figure 5 demonstrates a theorem at the college level, which Gemini can do both with and without *PhysLib* in context. From the comparison, we find that without the library, the proof is based solely on Mathlib. When the *PhysLib* is in the context, the proof tends to use the operations in the library and include explanatory comments. It indicates *PhysLib* can assist LLM's reasoning by providing a wider toolbox.

**Transfer tricks in mathematics** Figure 8 demonstrate a problem only solved by Goedel-Prover-V2-8B in our entire cycle of experiment. We conclude that this is a good transfer of following the NL intermediate steps and performing multiple trials learned in the math Lean training. From the NL statement demonstrated in Figure 7, we can observe that most steps are presented in the statement. Following these NL hints and many trials using `try` tactics, the Goedel-Provers can successfully solve the problem. It indicates that, although limited, the expert prover's capability of Lean math reasoning can be applied to formal physics reasoning.

**Why the general model performs better** To answer this question, we further analyze different topics solved by LLMs and find that general LLMs are typically better in thermo-dynamics. The Gemini successfully proved 22 theorems in the field, but DeepSeek-Prover-V2 only finished six theorems. We present one theorem that is proved by both Gemini-2.5 and DeepSeek-Prover-V2 to study the different behavior of their proof in Figure 6. We observe that the proof from the expert prover is more complex and tedious, while Gemini's proof is cleaner and more straightforward. Such a difference in proving consistently shows between the expert prover and general models. It suggests that, in fields underrepresented in training, the prolonged deliberation of expert models may lead to overthinking and suboptimal performance.

## 4 RELATED WORK

### 4.1 FORMAL REASONING

Formal Language (FL) reasoning involves expressing mathematical statements in a computer-verifiable manner. This approach mitigates ambiguity and provides a solid foundation for the reasoning process. Researchers have developed many FLs in the last decades, such as Isabelle (Paulson, 1994), CoQ (Coq, 1996), Metamath (Megill & Wheeler, 2019), and Lean (De Moura et al.,

2015). Among these, Lean4 (Moura & Ullrich, 2021) receives significant attention from the field due to its extensive foundation library of Mathlib (mathlib Community, 2020). A series of datasets and benchmarks has been developed to advance LLM Lean4 reasoning. For example, MiniF2F (Zheng et al., 2021) formalized competition-level math problems across multiple proof languages, ProofNet (Azerbayev et al., 2023) increases the level of difficulty to college-level, and PutnamBench (Gulati et al., 2024) serves as college-level competition problems, with MATPBench (He et al., 2025b) taking a step further by extending formal proof into multi-modal reasoning. Meanwhile, large-scale datasets have also been proposed to support LLM training, pushing the training data from 100k level (Wang et al., 2024a) to millions level (Lin et al., 2025a; Dong & Ma, 2025). At the same time, a series of Lean expert models are developed to perform formal math reasoning, representing works like DeepSeek-Prover Family (Xin et al., 2024; Ren et al., 2025), TheoremLlama (Wang et al., 2024a), Goedel-Prover Family (Lin et al., 2025a;b), LoT-Solver (Wang et al., 2025c), Kimina-Prover (Wang et al., 2025a), BFS-Prover (Xin et al., 2025), and GAR (Wang et al., 2025d). However, only a few methods (Wang et al., 2025b; Yao et al., 2025) tried to extend the formal reasoning beyond verifiable languages to the natural domain.

## 4.2 Lean in Subjects Beyond Mathematics

The application of Lean is gradually expanding from mathematics to a wider range of scientific fields, including chemical physics (Bobbin et al., 2024), molecular simulation (Ugwuanyi et al., 2025), and electrical engineering (blacksph3re, 2025). Some projects, such as Tooby-Smith (2025), attempt to formalize mechanics and high-energy physics using tools such as tensors and index notation. However, most of these studies are still limited to a small-scale, non-modular level for specific theorems, and lack a standardized evaluation system. Although the Lean expert provers perform well on mathematical benchmarks, their ability to transfer their reasoning skills to other fields, such as physics, remains to be examined.

## 4.3 Physics Datasets

In the field of physical reasoning research, the construction of data sets has long been a core issue. Physical problems have been extensively studied in the fields of natural language (NL) and machine learning, giving rise to several benchmarks that focus on physical understanding and reasoning. For instance, PhysBench (Chow et al., 2025) and PHYBench (Qiu et al., 2025) evaluate the physical reasoning ability of models; at the course level, UG-Physics (Xu et al., 2025) and the PHYSICS dataset (Zheng et al., 2025) provide thousands of textbook-style questions; at the competition level, OlympiadBench (He et al., 2024) includes 8,476 bilingual physics competition questions, HiPhO (Yu et al., 2025) covers the recent international physics olympiad questions, and PhysReason (Zhang et al., 2025) emphasizes the combination of multi-step reasoning and image understanding. Furthermore, CAMEL-Physics (Li et al., 2023) expanded the scale to tens of thousands of questions through automatic generation technology, providing a broader resource for model evaluation and training. Overall, these multi-level physical datasets laid the foundation for testing the physical reasoning ability of large language models under natural language conditions.

## 5 Conclusion

This paper presents Lean4PHYS, a comprehensive framework to support Lean4 physics reasoning. The framework includes *PhysLib*, an extensible, community-driven foundation library that sets the cornerstone for units, fields, and theorems for formal physics reasoning. To evaluate LLMs' performance on formal physics reasoning, we propose *LeanPhysBench*, which is, to the best of our knowledge, the first benchmark for Lean4-based physics reasoning. Based on the Lean4PHYS, we conduct extensive experiments to provide an overview of LLMs' performance on such tasks and the effectiveness of our *PhysLib*. We find that expert provers do not outperform large general models in most cases of formal physics reasoning. It indicates limited transfer capability from mathematical reasoning, despite the fact that they are all Lean4-based. Furthermore, the experiment shows that with *PhysLib* in the context, LLMs' performance on *LeanPhysBench* increases by **11.90%** on average. Our work establishes a general principle for extending the formalization of physics and other natural sciences beyond mathematics into a verifiable system.

## 6    DISCUSSION & FUTURE WORK

Lean4PHYS, to the best of our knowledge, the first work focusing on LLM-based formal physics reasoning, paves the way for several promising directions for future studies. First, as *PhysLib* grows, it may exceed the context length of current models. Thus, more sophisticated integration methods such as retrieval-augmented generation or tool-calling mechanisms will be necessary to select relevant theorems for the problem at hand. Second, since *PhysLib* is in its early stage with many fields and theorems yet to be implemented, developing autoformalization techniques to convert existing physics knowledge into Lean4 format and integrate it into *PhysLib* becomes vital. Third, our findings on limited cross-domain transfer suggest the need for unit-aware training strategies or adaptive reasoning techniques to bridge the gap between mathematical and physical formal reasoning. Furthermore, we acknowledge that the current benchmark primarily evaluates formal computation with physical units rather than deriving physical laws from first principles. Bridging this gap requires deeper integration between *PhysLib* and efforts such as PhysLean (Tooby-Smith, 2024). In conclusion, Lean4PHYS opens up many promising future directions for Lean4 formal physics reasoning.

## ACKNOWLEDGMENTS

We would like to express our gratitude to Yifei Xia for her valuable assistance in figure plotting and helpful comments during the paper-writing process. We thank Wenyuan Wang for sharing his precious experience in the engineering aspects of formal verification. We also thank Peng Chen for generously providing original collections of high school physics problems, which served as valuable references during the design phase of Lean4PHYS.

ETHICS STATEMENT

This work focuses on formalizing university physics problems in Lean4 and evaluating the formal physical reasoning capabilities of LLMs. To our knowledge, we strictly adhered to the ethical guidelines of the conference and considered the following aspects:

- **Data Sources:** All the physics problems are derived from publicly available university textbooks and competition problem collections. Appropriate citations have been provided, and no private or sensitive data has been used. In *LeanPhysBench*, the college-level statements are formalized from the physics problems in the university textbook (Young & Freedman, 2019) and the competition-level statements formalized from the physics Olympiad practice book (Shu et al., 1999).

- **Copyright Statement:** Our work only utilizes limited portions of the original copy of Young & Freedman (2019); Shu et al. (1999) for non-commercial research. Furthermore, our approach involves referencing short excerpts with proper attribution, a practice that does not substitute for original textbooks nor diminish their commercial value. As advised by professional legal counsel, our method of extracting underlying physics logic and creating the Lean4 benchmark does not constitute infringement and requires no explicit authorization. Nevertheless, we acknowledge that the interpretation of copyright law may vary across jurisdictions, and users of *LeanPhysBench* should use the benchmark in compliance with applicable laws. Twenty side-by-side comparisons of our formalized statements and the source materials are provided as supplementary material.

- **Licensing and Attribution:** The *PhysLib* is licensed with Apache-2.0, of which parts come from Tao (2025) , aligned with the origin repository. The *LeanPhysBench* is licensed with CC BY-NC 4.0, aligned with the copyright protection range of the source materials. Additionally, the *LeanPhysBench* may not be used to train, fine-tune, or evaluate any machine learning or AI models, regardless of whether the use is commercial or non-commercial.

- **Responsible Use of LLMs:** See A for details.

REPRODUCIBILITY STATEMENT

We have made efforts to ensure that our work is reproducible. The detailed description of the Lean4PHYS framework is presented in Section 2, with experiment details provided in Section 3 and Appendices B, D. We will open source the code of Lean4PHYS, consisting of a community-driven repository and a benchmark, at `https://github.com/ShirleyLIYuxin/Lean4PHYS`.

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

## A    USE OF LLMS

LLMs are used to assist in the development and verification of physics problems. Specifically, they generate detailed step-by-step solutions to help annotators better understand the reasoning process and calculation steps of the problems, as well as check the correctness and completeness of natural language answers. However, all the outputs were subsequently reviewed and verified by human annotators to ensure accuracy and reliability. Furthermore, LLMs are also utilized to enhance the clarity and readability of the manuscripts, including correcting grammar and sentence structures, as well as maintaining the consistency of chart titles with the layout of the tables.

## B    IMPLEMENTATION DETAIL

The generation configuration for the LLM roll-out of the experiment is as follows:

- Top-p: 0.95
- Temperature: 0.8
- Maximum tokens per generation: 16,384
- Repetition penalty: 1.0

Open-source models are tested under a 4-card H20 server. The entire open-source LLM roll-out process costs about 2 days. After the release of our initial draft, we found that two problems in "Comp-Hard" problems have unintended trivial shortcuts. We have fixed the problems to ensure the correctness of our benchmark. It caused the updated main experiment results in Section 3.2 to differ from those in our initial draft.

## C    COMPARISON WITH PHYSLEAN (TOOBY-SMITH, 2024)

This section aims to provide an explicit comparison between Lean4PHYS and PhysLean Tooby-Smith (2024), works that all aim to formalize physics reasoning in Lean4. We regard two works as highly complementary efforts of different perspectives and domains.

In the initial development stage of Lean4PHYS, we tried to formalize the problems based on the PhysLean repository. However, such initial attempts failed. This is because PhysLean excels at formalizing advanced theoretical physics (e.g., high-energy physics, quantum field theory) with sophisticated mathematical structures, such as fiber bundles and index notation. This design naturally differs from our target of developing Lean4PHYS, which focuses on foundational, college-level physics reasoning with an emphasis on creating a general-purpose infrastructure and a benchmark for LLM evaluation. The key distinction is that Lean4PHYS aims to establish a foundation for automated reasoning by LLMs, while PhysLean focuses on expert-driven formalization of advanced theories. In summary, we acknowledge the efforts of PhysLean (Tooby-Smith, 2024) in pioneering physics formalization in Lean4, while highlighting the unique role of Lean4PHYS in bridging physics formalization with LLM-based automated reasoning in formal physics of Lean4.

# D  LLM PROMPT TEMPLATES

We provide the prompt templates used to guide LLMs in generating Lean4 proofs.

**With *PhysLib*:**

```
Please first learn the new library besides mathlib and usage
↪  examples before answering the question. You should refer to
↪  the new unit system.

{PhysLib}

Complete the following Lean 4 code.
Provide your response in two parts, each enclosed in separate
↪  markdown code blocks:

```plan
# Proof Plan
- Outline the main proof steps and strategies.
- Highlight key intermediate lemmas and structures.
- Describe how to connect them to form the final proof.
```
```lean4
{Lean4_header}

/-- {NL_statement} -/
{Lean4_statement}
```
```

**Without *PhysLib*:**

```
Complete the following Lean 4 code.
Provide your response in two parts, each enclosed in separate
↪  markdown code blocks:

```plan
# Proof Plan
- Outline the main proof steps and strategies.
- Highlight key intermediate lemmas and structures.
- Describe how to connect them to form the final proof.
```
```lean4
{Lean4_header}

/-- {NL_statement} -/
{Lean4_statement}
```
```

# E  EXPERIMENT

## E.1  FORMAT STUDY

This appendix, based on the example shown in Figure 4, provides a more detailed analysis of the problem format described in Section 3.3, with a particular focus on the characteristics of the questions themselves.

Taking a college-level problem as an example, the upper left part of Figure 4 shows a force calculation problem involving two point charges on the x-axis. This problem involves various physical quantities, including charges, distances, and forces, and explicitly uses SI units such as nanocoulomb and m. The problem mainly relies on numerical calculations, and the magnitude of the force can be computed directly using Coulomb's law, with almost no need for multiple-step formula derivations. However, due to the potential for dimensional issues when combining different physical quantities in calculations, the solution to this problem requires maintaining unit consistency, which underscores the importance of the unit system in PhysLib for formal calculations. Such problems are typical of the "formula application type" and are suitable for verifying the model's ability in basic numerical reasoning and unit handling.

For the simple problem of the competition (Competition-Easy), the problem of the parallel plate capacitor shown in the lower left corner of Figure 4 is a typical example. The core formula of the problem is $C = q/V$, and only one or two steps of substitution are required for the calculation. The problem provides values for charge, voltage, and plate spacing and requires sequential substitution into the formula to complete the solution. Therefore, it involves combining numbers and symbols. This type of problem belongs to the "transitional derivation problem", similar to the intermediate math problems in Lean's MiniF2F, mainly testing the model's ability to perform algebraic operations and to substitute formulas. Compared to college-level problems, it has slightly increased logical reasoning requirements.

And the competition-hard problems (Competition-Hard), such as the pulley law problem shown in the right column of Figure 4, significantly increase the difficulty. The problem involves a continuous function of tension varying with angle and requires deriving the final result by integration or logarithmic relationships. Besides the multi-step formula derivation, the problem also involves relationships and function expressions among multiple physical quantities, requiring an understanding of the dependencies and combination rules among the model's symbols. Solving such issues not only requires multi-step logical analysis but also involves function operations and advanced symbolic reasoning, which are typically highly difficult, and testing the model's advanced physical reasoning ability.

### College

```
Two point charges are located on the x-axis
    of a coordinate system: q1 = 1.0 nC is
    at x = +2.0 cm, and q2 = -3.0 nC is at
    x = +4.0 cm. What is the total
    electric force exerted by q1 and q2 on
    a charge q3 = 5.0 nC at x = 0?
theorem Electromagnetism_3_University
    (q1 q2 q3 : Charge) (x1 x2 x3 : Length)
    (hq1 : q1 = SI.nano (1  · coulomb))
    (hq2 : q2 = SI.nano (-3 · coulomb))
    (hq3 : q3 = SI.nano (5  · coulomb))
    (hx1 : x1 = ((0.02:ℝ) · meter))
    (hx2 : x2 = ((0.04:ℝ) · meter))
    (hx3 : x3 = 0)(F   : Force)
    (hF : F = K * q3 * (q1 / (x1 - x3)^2 +
    q2 / (x2 - x3)^2)):
    F = ((9:ℝ)*(10:ℝ)^(-22:ℚ)/32) · newton
    := by
  simp [←Scalar.val_inj, hF, hq1, hq2,
    hq3, hx1, hx2, hx3, K, SI.nano,
    coulomb, meter, newton]
  norm_num
```

### Competition-Easy

```
The plates of a parallel-plate capacitor
    are 2.50 mm apart, and each carries a
    charge of magnitude 80.0 nC. The
    plates are in vacuum. The electric
    field between the plates has a
    magnitude of \(4.00 \times
    10^{6}\,\text{V/m}\). What is the
    capacitance?
theorem Ch13_electro_question_8
    (V:Voltage)(C:Scalar
    capacitance_unit)(q:Charge)
    (E:Scalar
    (force_unit-charge_unit))(d:Length)
    (h: capacitance_unit=charge_unit -
    voltage_unit)
    (hC:C=(q/V).cast h)(hq:q=SI.nano (80 ·
    coulomb))
    (hV:V=E*d)(hE:E=(4e6:ℝ) · StandardUnit
    _)
    (hd:d=SI.milli ((2.5:ℝ)· meter)):
    C=(1 / 125000000000:ℚ) · StandardUnit _
    := by
  have hC_expanded : C = (q/(E*d)).cast h :=
    by rw [hC, hV]
  rw [hC_expanded, hq, hE, hd]
  simp [nano, milli, coulomb, meter, ←
    Scalar.val_inj]
  norm_num
```

### Competition-Hard

```
Capstan law: If a rope of coefficient of
    friction μ wraps n turns (θ_total =
    2πn) around a post, the tension ratio
    between the heavy side M and the light
    side m satisfies n = (1 / (2πμ)) *
    log(M / m) assuming M > m > 0 and μ >
    0.

theorem Ch2_Q1
  (M m : Mass)
  (μ : ℝ)
  (n : ℝ)
  (θ_total : ℝ := 2 * Real.pi * n)
  (T : ℝ → Force)
  (h_pos : 0 < M.val ∧ 0 < m.val ∧ 0 < μ)
  (hM_gt_m : M.val > m.val)
  (T_light_def : T 0 = m * g)
  (T_heavy_def : T θ_total = M * g)
  (capstan_differential : ∀ θ : ℝ, deriv
    (fun θ' => (T θ').val) θ = μ * (T θ
    ).val)
  (capstan_integral : Real.log ((T θ
    _total).val / (T 0).val) = μ * θ_total)
  (theta_def : θ_total = 2 * Real.pi * n) :
  n = (1 / (2 * Real.pi * μ)) * Real.log
    (M.val / m.val) := by
  rcases h_pos with ⟨hM, hm, hmu⟩
  have h1 : Real.log ((T θ_total).val / (T
    0).val) =
            Real.log ((M * g).val / (m *
    g).val) := by rw [T_heavy_def,
    T_light_def]
  have h2 : Real.log ((M * g).val / (m *
    g).val) = Real.log (M.val / m.val) :=
    by
    have h3 : (M * g).val / (m * g).val =
    M.val / m.val := by
      field_simp; ring_nf; simp; ring
    rw [h3]
  have h3 : Real.log (M.val / m.val) = μ * θ
    _total := by linarith
    [capstan_integral, h1, h2]
  have h4 : Real.log (M.val / m.val) = μ *
    (2 * Real.pi * n) := by rw [theta_def]
    at h3; linarith
  have h5 : μ ≠ 0 := by linarith
  have h6 : Real.pi ≠ 0 := Real.pi_ne_zero
  have h7 : Real.log (M.val / m.val) = 2 *
    Real.pi * μ * n := by linarith [h4]
  have h8 : n = (Real.log (M.val / m.val))
    / (2 * Real.pi * μ) := by field_simp;
    linarith
  rw [h8]; field_simp; ring_nf; field_simp;
    ring
```

Figure 4: Three sampled physics questions from *College Textbook*, *Competition-Easy*, *Competition-Hard* problems. Each example shows the natural language problem statement followed by its corresponding Lean formalization with a verified proof.

### E.2 CASE STUDY

**Behavior of the same theorem with and without *PhysLib*** Figure 5 illustrates the different proof strategies employed by Gemini-2.5-pro when solving the same college-level mechanics problem, with and without *PhysLib*. Without *PhysLib*, the proof is relatively concise and almost entirely relies on the assumptions $ha$ and $hT$. For the acceleration $a$, the proof is completed by substituting the assumption into `rw [ha]` and then using basic simplification and multiplication/division commutative laws (`simp` and `mul_div_right_comm`) to transform the algebraic expression. This indicates that in the absence of domain-specific libraries, LLM relies on the general tactics of Mathlib to generate proofs, which are rather mechanical but can still accomplish the task correctly. When *PhysLib* is introduced, the proof becomes more structured and systematic. After substituting the assumption $a$, the proof utilizes the `Scalar.val_inj` lemma from PhysLib to transform the goal into an equation in the underlying real numbers, enabling LLM to handle algebraic operations of physical quantities more directly. Subsequently, the proof is completed by applying simplification (`simp`) and `ring` tactics to prove the algebraic identity. Overall, the proof with *PhysLib* demonstrates a richer use of tools and a more robust reasoning path.

**Transfer tricks in mathematics** During the process of solving the problem depicted in Figure 7 and Figure 8, Goedel-Prover demonstrated a variety of systematic proof strategies, which originated from its accumulated experience in mathematical Lean problems and were successfully transferred to formal proofs in physics. Firstly, the Prover would break down the overall proof goal into multiple sub-goals, such as proving $\mu_s = 0.46$ first, then proving $\mu_k = 0.40$, and finally combining them into a logical conjunction $(\mu_s = 0.46) \wedge (\mu_k = 0.40)$. This sub-goal decomposition strategy enables the Prover to handle complex problems step by step, with each sub-goal being independently verifiable and provable, thereby enhancing the reliability and success rate of automated proofs. This strategy is directly borrowed from the experience of handling complex theorems in mathematical Lean.

Within each sub-goal, the Prover will conduct a step-by-step verification of intermediate arithmetic or type values. For instance, it will first verify that `Scalar.val f_s_max = 230`, then `Scalar.val n = 500`, and subsequently calculate $(\mu_s : \mathbb{R}) = 0.46$ based on these intermediate results. This step-by-step verification method ensures that each stage is correct and error-free, preventing error accumulation and forming a reliable proof chain, which is exactly the same as the step-by-step verification of sub-conclusions in the Lean mathematical problems. When dealing with auxiliary steps such as type conversion or arithmetic simplification, Prover employs a multiple attempts strategy, such as using `try field_simp at *` or `try norm_cast at *`. These lightweight strategies enable Prover to automatically handle potential obstacles without requiring in-depth domain knowledge. The effectiveness of this strategy also stems from the experience of handling auxiliary steps through different tactics in Lean problems, achieving cross-domain transfer.

Finally, after completing the proofs of each sub-goal, the Prover uses the logical constructor to integrate the results, such as `exact ⟨h4, h7⟩`, combining the proofs of each sub-goal into an overall theorem. This method is a typical approach in mathematical Lean proofs and is also applicable in physical formalization problems, demonstrating the universality of the proof strategy and its ability for cross-disciplinary transfer.

**Why the general model performs better** We analyzed the proof style depicted in Figure 6 to explain why general-purpose models (such as Gemini-2.5-pro) can outperform expert theorem provers in under-trained domains. The theorem `Ch10_question_4` discussed involves calculating the thermodynamic index $k$ based on the known pressures $P_1, P_2$, volumes $V_1, V_2$, and temperatures $T_1, T_2$. Although both DeepSeek-Prover-V2 and Gemini-2.5-pro can complete the proof, their proof methods differ significantly. DeepSeek-Prover-V2 generates a highly step-by-step proof, gradually substituting and simplifying, explicitly introducing auxiliary steps such as $h_1$ and $h_2$, and repeatedly invoking simplification strategies for intermediate calculations; while Gemini-2.5-pro generates a concise proof by directly using all the assumptions at once and leveraging advanced simplification strategies to directly reduce the target to a numerical equation, omitting the lengthy intermediate steps.

From the perspective of the complexity of the steps, the proof of DeepSeek-Prover-V2 involves over twenty nested operations, while Gemini-2.5-pro can complete the proof through only two main

strategies. This indicates that although expert proofers can ensure the rigor of the form, in this type of domain that mainly relies on numerical calculations and has less symbolic reasoning, their step-by-step operations appear cumbersome and lengthy. Further analysis reveals that DeepSeek-Prover-V2 adopts a fine-grained approach, explicitly handling each substitution and simplification, while Gemini-2.5-pro conducts reasoning at a higher abstract level and uses the built-in simplification rules in the system to quickly complete the proof. In thermodynamic problems, higher-level abstraction can reduce unnecessary complex calculations and improve efficiency.

Furthermore, the expert prover mainly focuses on pure mathematical theorems during training. Therefore, in application fields such as thermodynamics, it is prone to introduce additional intermediate steps. This "overthinking" ensures rigor but may result in lengthy proofs and even increase the risk of errors in complex substitutions. In contrast, general large models possess flexible reasoning capabilities and can efficiently combine simplification rules with numerical calculations. In this case, Gemini-2.5-pro successfully mapped all hypotheses to numerical verification through a single `simp` call, demonstrating its adaptability to training deficiencies.

---

**Gemini-2.5-pro without *PhysLib***

```
theorem Mechanics_73_University
  (T : Force)(m_1 m_2 : Mass)(a : Acceleration)
  (ha : a = m_2 * g / (m_1 + m_2))
  (hT : T = (m_1 * m_2) / (m_1 + m_2) * g):
  (a = (m_2 / (m_1 + m_2)) * g ∧ T = (m_1 * m_2) / (m_1 + m_2) * g) :=
    by
  constructor
  · rw [ha]
    ext
    simp
    rw [mul_div_right_comm]
  · exact hT
```

---

**Gemini-2.5-pro with *PhysLib***

```
theorem Mechanics_73_University
  (T : Force)(m_1 m_2 : Mass)(a : Acceleration)
  (ha : a = m_2 * g / (m_1 + m_2))
  (hT : T = (m_1 * m_2) / (m_1 + m_2) * g):
  (a = (m_2 / (m_1 + m_2)) * g ∧ T = (m_1 * m_2) / (m_1 + m_2) * g) :=
    by
  constructor
  . -- The first goal is to prove the rearranged expression for
  acceleration `a`.
    -- We start by substituting `a` with its given definition from
  hypothesis `ha`.
    rw [ha]
    -- The goal is now to prove an algebraic equivalence between two
  `Scalar` expressions.
    -- We use the `Scalar.val_inj` lemma to transform this into an
  equality of their
    -- underlying real values. This is a standard technique in the
  PhysLib library.
    simp only [← Scalar.val_inj]
    -- Now, with the goal being an equality of real numbers, we can
  use `simp` to apply
    -- the library's rules for how `.val` distributes over arithmetic
  operations.
    simp
    -- The resulting goal is `m_2.val * g.val / (m_1.val + m_2.val) =
  m_2.val / (m_1.val + m_2.val) * g.val`.
    -- This is a standard algebraic identity which the `ring` tactic
  can solve.
    ring
  . -- The second goal is to prove the expression for tension `T`,
  which is identical
    -- to the hypothesis `hT`.
    exact hT
```

Figure 5: Two examples from *LeanPhysBench* demonstrating different proofs of the same college-level problem, generated by Gemini-2.5-pro with and without PhysLib.

**DeepSeek-Prover-V2**

```
theorem Ch10_question_4
    (P1 P2:Pressure) (V1 V2:Volume) (T1 T2:Temperature) (k:ℝ)
    (hV1:V1= 13· StandardUnit _)
    (hT1:T1=((1.8*15+273.15):ℝ) · kelvin)
    (hT2:T2=((-44.5*1.8+273.15):ℝ) · kelvin)
    (hP1:P1=((1.01*10^5):ℝ) · pascal)
    (hP2:P2= 868 · pascal)
    (hV2:V2=V1*T2*P1/(T1*P2))
    (hk:k=Real.rpow (V2/V1).val ((1/3):ℝ)):
    k=Real.rpow (10832250 / 144739) (1 / 3) := by
  have h_main : k = Real.rpow (10832250 / 144739) (1 / 3) := by
    have h₁ : k = Real.rpow (V2 / V1).val ((1 / 3 : ℝ)) := by rw [hk]
    rw [h₁]
    have h₂ : (V2 / V1).val = (10832250 / 144739 : ℝ) := by
      -- Simplify the expression for V2 / V1
      simp_all [Pressure, Volume, Temperature, add_assoc, add_comm,
    add_left_comm, mul_assoc, mul_comm, mul_left_comm,
    div_eq_mul_inv, mul_inv_cancel_left]
      <;> ring_nf at *
      <;> norm_num at *
      <;> field_simp [StandardUnit, mul_assoc, mul_comm,
    mul_left_comm] at *
      <;> ring_nf at *
      <;> norm_num at *
      <;> nlinarith
    rw [h₂]
    <;> simp [Real.rpow_def_of_pos]
    <;> norm_num
    <;> ring_nf
    <;> norm_num
    <;> field_simp
    <;> ring_nf
    <;> norm_num
    <;> nlinarith
  exact h_main
```

**Gemini-2.5-pro**

```
theorem Ch10_question_4
    (P1 P2:Pressure)(V1 V2:Volume)(T1 T2:Temperature)(k:ℝ)
    (hV1:V1= 13· StandardUnit _)
    (hT1:T1=((1.8*15+273.15):ℝ) · kelvin)
    (hT2:T2=((-44.5*1.8+273.15):ℝ) · kelvin)
    (hP1:P1=((1.01*10^5):ℝ) · pascal)
    (hP2:P2= 868 · pascal)
    (hV2:V2=V1*T2*P1/(T1*P2))
    (hk:k=Real.rpow (V2/V1).val ((1/3):ℝ)):
    k=Real.rpow (10832250 / 144739) (1 / 3) := by
  -- Use all hypotheses to substitute the variables and simplify the
    expression.
  -- `simp` will use `hk` to substitute `k`, `hV2` to substitute `V2`,
  -- and the other hypotheses to substitute the numerical values.
  -- The `@[simp]` lemmas in the UnitsSystem library will handle the
    `.val` operations.
  simp [hk, hV2, hV1, hT1, hT2, hP1, hP2]
  -- The goal is now a purely numerical equality. Use `norm_num` to
    verify it.
  norm_num
```

Figure 6: Comparison of Proof Styles by DeepSeek-Prover-V2 and Gemini-2.5-pro.

Goedel-Prover with *PhysLib* (nature statement)

Question 74: You want to move a 500 N crate across a level floor. To start the crate moving, you have to pull with a 230 N horizontal force. Once the crate starts to move, you can keep it moving at constant velocity with only 200 N. What are the coefficients of static and kinetic friction? Follow the process: Just before the crate starts to move, we have:

$$\sum F_x = T + \big( -(f_s)_{\text{max}} \big) = 0 \quad \text{so} \quad (f_s)_{\text{max}} = T = 230 \text{ N}$$

$$\sum F_y = n + (-w) = 0 \quad \text{so} \quad n = w = 500 \text{ N}$$

Now we solve $(f_s)_{\text{max}} = \mu_s n$, for the value of $\mu_s$:

$$\mu_s = \frac{(f_s)_{\text{max}}}{n} = \frac{230 \text{ N}}{500 \text{ N}} = 0.46$$

After the crate starts to move, we have:

$$\sum F_x = T + \big( -f_k \big) = 0 \quad \text{so} \quad f_k = T = 200 \text{ N}$$

$$\sum F_y = n + (-w) = 0 \quad \text{so} \quad n = w = 500 \text{ N}$$

Using $f_k = \mu_k n$, we find:

$$\mu_k = \frac{f_k}{n} = \frac{200 \text{ N}}{500 \text{ N}} = 0.40$$

show that the coefficients of static is $\mu_s = 0.46$ and kinetic friction is $\mu_k = 0.40$.

Figure 7: Natural Language Statement of a Problem Solved Exclusively by Goedel-Prover.

---

**Goedel-Prover with *PhysLib* (Lean4 statement)**

```
theorem Mechanics_74_University_0
  (f_s_max f_k n w : Force)(μ_s μ_k : ℚ)
  (hw : w = 500 · newton)
  (hn : n = w)
  (hf_s_max : f_s_max = 230 · newton)
  (hf_k : f_k = 200 · newton)
  (hμ_s : μ_s = Scalar.val f_s_max / Scalar.val n)
  (hμ_k : μ_k = Scalar.val f_k / Scalar.val n):
  (μ_s = (0.46 : ℚ) ∧ μ_k = (0.40 : ℚ)) := by
  have h1 : Scalar.val f_s_max = 230 := by
    rw [hf_s_max]
    simp [Scalar.val_smul, newton]

  have h2 : Scalar.val n = 500 := by
    have h2₁ : n = 500 · newton := by
      rw [hn, hw]
    rw [h2₁]
    simp [Scalar.val_smul, newton]

  have h3 : (μ_s : ℝ) = (0.46 : ℝ) := by
    have h3₁ : (μ_s : ℝ) = Scalar.val f_s_max / Scalar.val n := by
      norm_cast at hμ_s ⊢
    rw [h3₁]
    rw [h1, h2]
    norm_num

  have h4 : μ_s = (0.46 : ℚ) := by
    have h4₁ : (μ_s : ℝ) = (0.46 : ℝ) := h3
    norm_num at h4₁ ⊢
    norm_num
    (try field_simp at *)
    (try norm_cast at *)

  have h5 : Scalar.val f_k = 200 := by
    rw [hf_k]
    simp [Scalar.val_smul, newton]

  have h6 : (μ_k : ℝ) = (0.40 : ℝ) := by
    have h6₁ : (μ_k : ℝ) = Scalar.val f_k / Scalar.val n := by
      norm_cast at hμ_k ⊢
    rw [h6₁]
    rw [h5, h2]
    norm_num

  have h7 : μ_k = (0.40 : ℚ) := by
    have h7₁ : (μ_k : ℝ) = (0.40 : ℝ) := h6
    norm_num at h7₁ ⊢
    (try field_simp at *)
    (try norm_cast at *)

  have h8 : (μ_s = (0.46 : ℚ) ∧ μ_k = (0.40 : ℚ)) := by
    exact ⟨h4, h7⟩

  exact h8

end Mechanics
```

Figure 8: Lean4 Formalization and Proof of the Problem Shown in Figure 7 by Goedel-Prover.

