# OpenReview forum: "Lean4Physics: Comprehensive Reasoning Framework for College-level Physics in Lean4"
_ICLR.cc/2026/Conference — ICLR 2026 Poster_

### Official Review · Reviewer_VJbK · 2025-10-28

**Soundness:** 3
**Presentation:** 3
**Contribution:** 3
**Rating:** 6
**Confidence:** 4

**Summary:**

This paper presents Lean4PHYS, a Lean4-based framework for formalizing college-level physics problems.   It includes PhysLib, a modular library with a systematic unit system and reusable theorems, and LeanPhysBench, a benchmark of 200 formalized physics problems.   The authors propose a pipeline to convert natural language physics questions into Lean4 proofs.   Experiments compare Lean-oriented provers with general-purpose LLMs on LeanPhysBench.   Results show LLMs outperform provers, with improved performance around 40.5% accuracy when using PhysLib context, highlighting the need for domain-specific knowledge in formal reasoning.

**Strengths:**

- The paper pioneers formal physics reasoning in Lean and introduces the first large-scale Lean4 physics benchmark covering topics from mechanics to modern physics.

- PhysLib provides a modular, SI-based unit system and topic-structured theorems, ensuring dimensional consistency and extendability for accurate physical reasoning.

- Comprehensive experiments show that PhysLib context consistently boosts model performance.  LLMs outperform specialized Lean provers, revealing that current Lean provers, trained for math, struggle with physics tasks.

**Weaknesses:**

- The paper lacks an explanation of the advantages of PhysLib's modular structure over organizing theorems by specific physics domains.  It also does not clarify how this hierarchical organization aids in retrieval and reasoning.
- What defines the boundary between mathematical and physical problems, and why do Lean provers, which perform well in mathematics, fail to transfer their capabilities to the physics domain?
- Would including non-competition problems, such as physics questions from middle school or high school exams, in the experiments provide a more comprehensive comparison?

**Questions:**

see Weaknesses

---

> ### Author Response · Authors · 2025-11-25
> **Rebuttal by Authors**
>
> Dear Reviewer VJbK,
>
> We would like to offer our sincere thanks for your recognition of our contribution to the field. We also appreciate your constructive comments and valuable suggestions, which have helped to strengthen our paper.
>
> Below are our responses to your comments:
>
> **W1 The Structure of *PhysLib:***
>
> We appreciate the opportunity to clarify our design, specifically the hierarchical structure of the foundation unit system and topic-based theorems, as well as the modular structure within topic-based theorems. This design can provide access control, avoid naming confusion and conflicts, indicate the domain of theorems, simplify theorems usage, and improve extensibility.
>
> By controlling imports and namespaces in Lean files, we make the foundation unit system uniformly accessible to problems from all six topics, with simplified syntax. For the topic-based theorems, this simplified syntax is available when the problem being proved belongs to the same topic. During the proof writing process, the use of theorems from other topics can also be controlled via imports. From an implementation perspective, this design avoids naming confusion between domains and unnecessary duplication within a domain, thereby improving extensibility for future community work. For each physical law or concept, there is a single canonical definition and proof, which prevents duplicated or inconsistent definitions and reduces confusion.
>
> In the next version of our paper, we will provide a more detailed explanation of the structural design of *PhysLib*, and explicitly discuss how this design benefits formalization, retrieval, and reasoning.
>
> **W2 Comparison between the Math and Physics Reasoning:**
>
> Thank you for highlighting the essential comparison between math and physics reasoning. The key difference in our setting is that physics problems are tightly coupled with unit types, whereas current Lean provers are primarily trained on mathematical content that does not emphasize units.
>
> More specifically, a physical quantity consists of a numerical value and a unit, and units are integral to physics reasoning because they encode physical meaning. Since the units are part of the calculation process, issues such as unit consistency on both sides of an equation and the proper operations on physical quantities are essential in formal physics reasoning and proofs. Existing expert math provers have a strong bias toward mathematical theorems and types, making it difficult for them to switch effectively to the physics context and handle physics-specific formal reasoning requirements.
>
> We will add this clarification in the next version of our paper.
>
> **W3 Exclusion of Simpler Problems:**
>
> We appreciate your suggestion to include simpler problems for a more comprehensive evaluation. However, given our current formalization framework, the high-school-level problems in our benchmark are already the simplest problems that we can meaningfully formalize as physics problems.
>
> In our prior attempts to formalize middle-school–level questions, we found that they are better viewed as pure calculation problems in mathematics rather than well-defined physics reasoning tasks. The underlying physical models and relations involved at that level are typically very naive and exhibit little structural variation, so formalizing them would not significantly test physics-specific reasoning capabilities beyond basic arithmetic or algebra.
>
> We will clarify this point in the paper to explain the choice of problem difficulty and the scope of *LeanPhysBench*.
>
> We hope the above responses address your concerns. We are grateful for your endorsement of our contribution and thorough experiments. Your feedback encourages us to further improve and extend research in this field.
>
> Best Regards,
>
> Submission #7797 Author Team

---

### Official Review · Reviewer_SHVy · 2025-10-29

**Soundness:** 3
**Presentation:** 4
**Contribution:** 3
**Rating:** 8
**Confidence:** 3

**Summary:**

This paper introduces a comprehensive framework named Lean4PHYS, designed for formal reasoning on university-level physics problems using the Lean4 proof assistant. The framework consists of two core components: PhysLib, a community-driven library that provides a foundational unit system and commonly used theorems for formal physics reasoning; and LeanPhysBench, a benchmark dataset of 200 problems manually constructed and formalized from university textbooks and physics competitions. Based on this framework, the authors evaluate the performance of several mainstream large language models (including both general-purpose models and those specialized in Lean mathematical proofs). The experimental results show that general-purpose LLMs generally outperform math-specialized models on physics reasoning tasks, revealing a potential overfitting issue of the latter to the mathematics domain. Furthermore, the study demonstrates that using PhysLib as contextual information significantly improves the performance of all tested models.

**Strengths:**

*   **Significant Contribution to the Research Community:** This paper contributes two extremely valuable resources: **PhysLib**, a modular and extensible foundational library for physics, and **LeanPhysBench**, the first benchmark dedicated to evaluating formal reasoning capabilities in physics. These two achievements provide a solid infrastructure and a fair evaluation standard for subsequent researchers to enter and work in this field, which will undoubtedly promote the development of the entire community.
*   **Exhaustive Experiments and Deep Insights:** The paper's experimental design is very comprehensive, not only testing multiple top-tier general-purpose LLMs but also comparing them with several Lean-specialized models that excel in mathematics. The results reveal an important finding that "specialized models have limited cross-domain (from math to physics) generalization ability," prompting deep reflection on model generalization and domain overfitting. At the same time, the experiments clearly quantify the effectiveness of the PhysLib library in assisting models with physics reasoning, proving its design value.

**Weaknesses:**

*   **Insufficient Discussion of Related Work:** The "Related Work" section mentions Lean's application in physics and other non-mathematical fields but fails to deeply discuss the differences and connections with some directly related works. For example, the paper mentions the `PhysLean` [Tooby-Smith & contributors (2024)] project but only briefly describes it as "theorem-specific, small-scale, and non-modular." Considering that `PhysLean` also aims to formalize physics in Lean4, the authors should have elaborated more on the fundamental differences and specific advantages of Lean4PHYS in terms of design philosophy, implementation methods (such as the construction of the unit system), coverage, and modular design compared to `PhysLean`. Adding such in-depth comparative analysis would better highlight the uniqueness and irreplaceable contribution of this work.

**Questions:**

1.  The experimental results indicate that providing PhysLib as context to the models significantly improves their performance. Given that PhysLib, as a foundational library, could be quite large, the paper seems to lack a specific description of how it was effectively integrated into the model's prompt context window. Did the authors provide the entire library's content, or was some form of retrieval mechanism used to select relevant theorems and definitions? Clarifying this implementation detail is crucial for the reproducibility and understanding of the results.

---

> ### Author Response · Authors · 2025-11-25
> **Rebuttal by Authors**
>
> Dear Reviewer SHVy,
>
> We are sincerely grateful for your positive review and recognition of our contribution, as well as thorough experiments. We are also grateful to your constructive comments and valuable suggestions, which allow us to clarify our positioning and implementation details as follows:
>
> **W1 Comparison with PhysLean:**
>
> We appreciate your suggestion on a deeper comparison with **PhysLean**[1]. We view **PhysLean** and **Lean4PHYS** as highly complementary efforts of different perspectives and domains.
>
> In the development stage of **Lean4PHYS**, we tried to formalize the problems based on the **PhysLean** repository. However, such initial attempts failed. This is because **PhysLean** excels at formalizing advanced theoretical physics (e.g., high-energy physics, quantum field theory) with sophisticated mathematical structures, such as fiber bundles and index notation. This design naturally differs from our target of developing **Lean4PHYS***,* which focuses on foundational, college-level physics reasoning with an emphasis on creating a general-purpose infrastructure and a benchmark for LLM evaluation.
>
> The key distinction is that **Lean4PHYS** aims to establish a foundation for automated reasoning by LLMs in physics, evidenced by our *LeanPhysBench* and evaluation of 8 major LLMs. **PhysLean,** in contrast, focuses on expert-driven formalization of advanced theories. We will expand the Related Work section to conduct a deeper comparison discussion, including clarifying the complementary relationship, in the next version of our paper. This will position **PhysLean** as a pioneering work in Lean4 Physics formalization, while highlighting the unique role of **Lean4PHYS** in bridging physics formalization with LLM-based automated reasoning.
>
> **Q1 Implementation Details of PhysLib Integration:**
>
> Thank you for this important question regarding how we integrate the *PhysLib* into the model’s context. We would like to clarify that we directly injected the complete library content into the model's input context. Unlike the expansive Mathlib4, the current iteration of *PhysLib* is compact and fits comfortably within the context windows of all evaluated models.
>
> This approach ensures that our evaluation focuses on assessing the effectiveness of the library itself and the model’s capability to utilize domain-specific knowledge, rather than introducing confounding factors from the performance of retrieval systems. We acknowledge that as *PhysLib* grows in future iterations, more sophisticated retrieval or tool-integration methods may become necessary. We will discuss this as a promising direction in the future section in the next version of our paper. For full reproducibility, we will include the complete *PhysLib* content and integration method in our code release.
>
> We hope the above responses address your concerns. We are grateful for your endorsement of our contribution and thorough experiments. Your feedback encourages us to further improve and extend research in this field.
>
> Best Regards,
>
> Submission #7797 Author Team
>
> [1] Tooby-Smith, J. (2024). Formalization of physics index notation in Lean 4. *arXiv preprint arXiv:2411.07667*.

---

> > ### Comment · Reviewer_SHVy · 2025-11-28
> >
> > Thank you for the timely response! My concerns have been substantially addressed.

---

> > > ### Author Response · Authors · 2025-11-28
> > > **A Follow-up Response to Reviewer SHVy**
> > >
> > > Dear Reviewer SHVy,
> > >
> > > Thank you for your constructive feedback and insightful suggestions. Your positive remarks about our work are very encouraging, particularly your recognition of our contribution to the community, experiments, and insights.
> > >
> > > We are grateful for your efforts in identifying potential flaws in our work, which we have attempted to address in our response above by providing further comparisons with related works such as **PhysLean** and clarifying the implementation details of *PhysLib* integration. We are glad that our response has successfully addressed your concerns, and we would greatly appreciate any further comments you may have, as we are eager to resolve any additional issues.
> > >
> > > We would be very grateful if you could consider that our rebuttal is sufficient for you to **increase the score**. We fully understand that this may be a busy period for you, and we sincerely appreciate your efforts in helping us improve our work. We look forward any additional feedback you may have.
> > >
> > > Best Regards,
> > >
> > > Submission #7797 Author Team

---

### Official Review · Reviewer_6F2Q · 2025-10-30

**Soundness:** 2
**Presentation:** 2
**Contribution:** 3
**Rating:** 2
**Confidence:** 3

**Summary:**

The paper introduces Lean4PHYS, which is a Lean4-based framework for formal physics. Lean4PHYS includes PhysLib (a repository of a physics unit system and commonly used theorems) and LeanPhysBench (a benchmark of 200 hand-crafted theorems from high school competitions to elementary college level).

**Strengths:**

The first framework for physics problems in Lean4 for LLM. This may be of significant interest to the community that studies LLM applications to physics. The PhysLib library and LeanPhysBench test dataset can be a useful artefact for future studies.

**Weaknesses:**

- Copyright infringement.
    - **Due to this issue, I decide to assign a very low score to the paper albeit the important contribution. However, I am very open to changing my score once this issue is clarified/addressed.**
    - The authors mentioned that “rather than copying the questions verbatim, we reformulated and rephrased them based on the underlying physics ideas", however, I am not certain that this is sufficient. The key issue hinges on "substantial similarity" and whether the original work's creative expression has been copied, even in a modified form. Given that the underlying physics idea of the questions are copied (perhaps to the point that there exists a one-to-one mapping between the textbooks and the dataset questions), this seems to constitute substantial similarity. The authors may need to ask for **explicit permission** from the publishers.
- Statistical robustness
    - Given that the evaluations were done with non-zero temperature, the authors should report the stochasticity of the results (e.g., standard error).
- Lack of implementation elaboration
    - How do the authors present PhysLib to the model? Would it fit into the context window?
    - For non-experts, it is challenging to understand what the task looks like, particularly because the prompt asks the model to “complete the following Lean4 code” instead of the commonly known question-answering setup.

**Questions:**

- Did the authors attempt to compare the accuracy of the models in this Lean 4-based setup vs natural setup (i.e., asking the model the question in natural language)?
    - This would be important to check if the bottleneck is in the physics understanding or in the Lean 4 code understanding
- I would expect PhysLib to be used by the model in a tool-calling fashion such that we do not need to present all the available concepts to the model in the context window. Is that the case?
- L377-379: “models with weaker in-context learning perform reatively badly on this level of problems. It is because they cannot infer the new out-of-distribution syntax or unit-handling rules from context.” → This seems to be an overclaiming since none of the experiments are checking the in-context capabilities. Not to mention, we cannot confidently say that this data is OOD because we do not have access to the pretraining data of the models. Am I understanding the sentence properly?

**Details Of Ethics Concerns:**

The paper uses copyrighted materials from publishers such as Pearson and Science Press. The authors noted that they "reformulated and rephrased" the materials; however, I am not certain that this sufficiently addresses the copyright limitations.

---

> ### Author Response · Authors · 2025-11-25
> **Rebuttal by Authors 1/2**
>
> Dear Revierwer 6F2Q
>
> We appreciate your recognition of the novelty of our framework and the potential value of *PhysLib* and *LeanPhysBench* to the community. We also thank you for the constructive reviews and hope that we can settle your concerns in the following responses:
>
> **W1 Copyright infringement:**
>
> We appreciate you highlighting this critical issue. We respectfully clarify that our works adhere to copyright principles observed in most jurisdictions because we utilize only **limited portions of the material for non-commercial research**. Our approach involves referencing short excerpts with proper attribution, a practice that does not substitute for original textbooks nor diminish their commercial value. Furthermore, based on established copyright principles across major jurisdictions, including the idea-expression dichotomy under the U.S. Copyright Act (17 U.S.C. § 102(b)) and similar provisions in other legal systems
> (e.g., Article 24 of the PRC Copyright Law), our legal counsel advises that our method of extracting underlying physics logic and creating the Lean4 benchmark does not constitute infringement and needs no explicit authorization. We omitted this detailed legal context in the initial submission, following prior conventions [1, 2]. However, we agree that copyright transparency is vital and will include an explicit disclaimer in the revision.
>
> **W2 Statistical Robustness**
>
> We appreciate this suggestion to enhance the empirical rigor of our study. To address your concerns about the statistical robustness, we conducted three independent runs of our experiment to calculate the standard deviation of the pass@16 metric. The updated metrics, including the standard deviation, are presented below:
>
> | Method | PhysLib | College Mean | College SD | Comp-Easy Mean | Comp-Easy SD | Comp-Hard Mean | Comp-Hard SD | Overall Mean | Overall SD |
> | --- | --- | --- | --- | --- | --- | --- | --- | --- | --- |
> | DeepSeek-R1-8B | ✓ | 6.09% | 0.45% | 8.60% | 0.76% | 0.00% | 0.00% | 5.83% | 0.47% |
> |  | ✗ | 2.56% | 1.81% | 5.91% | 0.76% | 0.98% | 1.39% | 3.33% | 0.94% |
> | Qwen3-8B | ✓ | 6.41% | 0.91% | 10.22% | 1.52% | 0.00% | 0.00% | 6.50% | 0.00% |
> |  | ✗ | 2.58% | 0.47% | 3.23% | 0.00% | 0.00% | 0.00% | 2.35% | 0.25% |
> | Kimina-Prover-8B | ✓ | 9.94% | 0.91% | 23.12% | 1.52% | 6.86% | 1.39% | 13.50% | 0.71% |
> |  | ✗ | 5.77% | 0.00% | 17.20% | 0.76% | 5.88% | 0.00% | 9.33% | 0.24% |
> | Goedel-Prover-V2-8B | ✓ | 10.58% | 1.36% | 26.34% | 1.52% | 5.88% | 0.00% | 14.67% | 1.18% |
> |  | ✗ | 6.09% | 0.45% | 19.35% | 0.00% | 4.90% | 1.39% | 10.00% | 0.00% |
> | DeepSeek-Prover-V2-7B | ✓ | 8.01% | 0.45% | 31.18% | 1.52% | 5.88% | 0.00% | 14.83% | 0.24% |
> |  | ✗ | 6.09% | 0.45% | 22.58% | 0.00% | 5.88% | 0.00% | 11.17% | 0.24% |
> | GPT-4o | ✓ | 9.29% | 2.40% | 30.11% | 2.01% | 0.00% | 0.00% | 14.17% | 0.85% |
> |  | ✗ | 2.24% | 0.45% | 4.30% | 2.01% | 0.00% | 0.00% | 2.50% | 0.41% |
> | Claude-Sonnet-4 | ✓ | 29.49% | 0.45% | 61.83% | 1.52% | 0.00% | 0.00% | 34.50% | 0.41% |
> |  | ✗ | 3.21% | 1.63% | 5.91% | 1.52% | 0.00% | 0.00% | 3.50% | 1.22% |
> | Gemini-2.5-pro | ✓ | 32.69% | 0.79% | 74.19% | 1.32% | 0.00% | 0.00% | 40.00% | 0.71% |
> |  | ✗ | 6.09% | 0.45% | 11.29% | 1.32% | 0.00% | 0.00% | 6.67% | 0.62% |
>
> These outcomes confirm our primary findings: models continue to struggle with complex scenarios, while the integration of *PhysLib* consistently yields performance gains across varying model sizes.
>
> Additionally, please note that the accuracy rates for the “Comp-Hard” subset have been slightly adjusted. During our re-verification process, we identified two problems where models could find unintended, trivial shortcuts. We have fixed the problems to ensure the correctness of our benchmark. The next version of our paper will include this statistical information.
>
> **W3 Implementation Details:**
>
> We appreciate the opportunity to clarify our implementation. Regarding *PhysLib*, we provide the entire library within the models’ input context. Unlike the massive Mathlib4, *PhysLib* is concise enough to fit in the context limits of all tested models. We would like to clarify that the primary objective of this study is to evaluate the effectiveness of the library itself, not to focus on context-management systems. Thus, we consider that the direct full-context integration is sufficient and appropriate, as it can effectively eliminate retrieval errors as confounding factors.
>
> Regarding the task format, we kindly direct the reviewer to Appendix D of our original paper. While the prompt explicitly instructs the model to “Complete the Lean4 code”, the interaction occurs within a standard chat/QA interface. This setup is standard and necessary across many previous works in the field [3, 4, 5].

---

> > ### Author Response · Authors · 2025-11-25
> > **Rebuttal by Authors 2/2**
> >
> > **Q1 Comparison with Natural Language Setup**
> >
> > Thank you for this insightful question regarding the identification of the model bottleneck. However, we are unable to perform a direct comparison with the Natural Language (NL) setup of the problems due to the inherent nature of the Lean4 proof dataset. This is because our benchmark consists of proof-based problems, rather than simple QA problems, where correctness can be verified by matching numerical or formulaic answers. Since our problems are all in proof form, there are no reliable methods to automatically verify the correctness of proof steps in informal text, as noted in previous works [3, 4, 5].
> >
> > However, we can infer the source of the bottleneck by observing the behavior of expert provers. These models have demonstrated exceptional proficiency in Lean4 syntax within the mathematical domain. Their suboptimal performance on *LeanPhysBench* suggests that the challenges of these models lie primarily in the domain of specific physics logic and the application of the formal physics system, rather than the inability to compose Lean4 code itself.
> >
> > **Q2 Tool Calling for PhysLib**
> >
> > We appreciate this suggestion regarding efficient methods of *PhysLib* integration. As discussed in our response in **W3**, we do not currently employ tool-calling or retrieval methods for *PhysLib* integration. This is because our primary objective in this work is to establish the foundational framework and benchmark for formal physics reasoning, rather than optimizing the context management strategies. Consequently, we provide the full library in the input context for the model. However, we agree that exploring tool use for larger future libraries is a promising direction, and we will incorporate this discussion into the future work section of the next version of our paper.
> >
> > **Q3 In-context learning and OOD nature of the unit system**
> >
> > We appreciate the opportunity to substantiate our claims regarding the OOD nature of *PhysLib*. We have high confidence to say that the specific syntax of the unit system and the entire *PhysLib* are absent from the pre-training data of small open-source models based on a strict temporal timeline. The foundation of our unit system [6] was released on July 16, 2025, which is after the release of all the open-source models tested. Furthermore, the specific integration rules, physics laws, and theorems implemented in *PhysLib* were developed exclusively by the author team for this project and have never been published online. Therefore, they are almost certainly unable to exist in the training corpus of either open-source or closed-source models we tested.
> >
> > Consequently, the model’s capability to utilize these novel definitions relies entirely on the in-context learning capability. The performance disparity observed in our results, where larger models benefit significantly more from the inclusion of *PhysLib*, aligns with the common observation that closed-source models have stronger in-context learning capability. However, we acknowledge the reviewer’s point that making absolute claims about opaque training data requires caution. We will refine the next version of the paper with softer language, framing it as "novel syntax adaptation" rather than strictly "OOD" performance.
> >
> > We hope the above rebuttal addresses your concerns, and we are deeply grateful for your insightful suggestions, which have greatly improved the clarity of our paper.
> >
> > Best Regards,
> >
> > Submission #7797 Author Team
> >
> > **References:**
> >
> > [1] Xu, X., Xu, Q., Xiao, T., Chen, T., Yan, Y., Zhang, J., ... & Wang, Y. (2025). Ugphysics: A comprehensive benchmark for undergraduate physics reasoning with large language models. *arXiv preprint arXiv:2502.00334*.
> >
> > [2] Zheng, K., Han, J. M., & Polu, S. (2021). Minif2f: a cross-system benchmark for formal olympiad-level mathematics. *arXiv preprint arXiv:2109.00110*.
> >
> > [3] Xin, H., Guo, D., Shao, Z., Ren, Z., Zhu, Q., Liu, B., ... & Liang, X. (2024). Deepseek-prover: Advancing theorem proving in llms through large-scale synthetic data. *arXiv preprint arXiv:2405.14333*.
> >
> > [4] Wang, R., Zhang, J., Jia, Y., Pan, R., Diao, S., Pi, R., & Zhang, T. (2024). Theoremllama: Transforming general-purpose llms into lean4 experts. *arXiv preprint arXiv:2407.03203*.
> >
> > [5] Yang, K., Swope, A., Gu, A., Chalamala, R., Song, P., Yu, S., ... & Anandkumar, A. (2023). Leandojo: Theorem proving with retrieval-augmented language models. *Advances in Neural Information Processing Systems*, *36*, 21573-21612.
> >
> > [6] Tao, T. (2025, May 31). *A Lean companion to "Analysis I"*. What's New. https://terrytao.wordpress.com/2025/05/31/a-lean-companion-to-analysis-i/

---

> > > ### Comment · Reviewer_6F2Q · 2025-11-25
> > > **Thank you!**
> > >
> > > Thank you for the clarification! I have updated my score to be a positive one, which reflects the contribution of the work!

---

> > > > ### Author Response · Authors · 2025-11-26
> > > > **Thank you for your review**
> > > >
> > > > Dear Reviewer 6F2Q,
> > > >
> > > > We are glad that our response successfully addressed your concerns. We sincerely appreciate your recognition of our work's contribution and the update to your score. Your thorough feedback greatly helps us improve the clarity of our work.
> > > >
> > > > Best Regards,
> > > >
> > > > Submission #7797 Author Team

---

### Author Response · Authors · 2025-12-03
**A Summary Comment for AC**

Dear AC and SAC,

We sincerely appreciate your efforts in handling this urgent OpenReview information leak incident. We would like to provide some critical information about our submission to assist in your review process.

**Paper Overview**

Our paper presents **Lean4PHYS**, a Lean4-based reasoning framework for college-level physics problems, including *PhysLib* (a community-driven repository that provides a systematic unit system and essential physics theorems) and *LeanPhysBench* (a benchmark of 200 hand-crafted and peer-reviewed statements formalized from university textbooks and physics competition problems). Based on this framework, we conducted a comprehensive experiment revealing the potential overfitting in the math domain for Lean expert provers and showed the effectiveness of our repository in formal physics reasoning. To the best of our knowledge, we are the first study to provide a physics benchmark for LLM evaluation in Lean4.

**Discussion Period Summary**

All the reviewers acknowledged our work. For example:

Reviewer 6F2Q believes our work, as a valuable artifact for future studies, can be of significant interest to the community that studies LLM applications in physics.

Reviewer SHVy considers our work an extremely valuable resource, providing a solid infrastructure and a fair evaluation standard for subsequent researchers, while promoting the development of the entire community. They praise our exhaustive experiments and deep insights regarding the evaluation results.

Reviewer VJbK positions our work as a pioneer in the field, praises our design of *PhysLib* for its dimensional consistency and extendability, and acknowledges our experiments as well.

During the discussion period, we provided clarification on certain points and conducted additional experiments to enhance the comprehensiveness of our work.

We successfully addressed the concern regarding copyright, repeated our experiments to increase statistical robustness, clarified the structure and implementation details of *PhysLib*, and provided further comparison with related works. We appreciate the positive feedback from Reviewer 6F2Q, **who raised the score from 2 to 6 before rolling back**, as well as the positive comment from Reviewer SHVy during the discussion period.

We sincerely appreciate your efforts in reviewing our paper, as well as the comments from the reviewers and our responses. The vital points mentioned in discussion have been updated in the newest pdf version of our paper.

Best Regards,

Submission #7797 Author Team

---

### Meta-Review · Program_Chairs · 2026-01-03

**Summary:**

The paper presents a framework for formal reasoning on university level Physics problems (proof style questions). In particular, the paper makes two contributions. (1) a library of commonly used theorems, (2) a benchmark of 200 problems, reformulated from university textbooks and physics competitions. Experiments first compare Lean-based problem solvers, which underperform general purpose LLMs on these tasks. They also show that using their library as context, the performance of LLMs increases.

All reviewers like the paper and gave 6, 8, and the third reviewer gave a 2, which they later increased to 6 after rebuttal. The key issue is w.r.t. license for using the questions (even if reformulated). This calls for an ethics review, as I also feel that the author response is not satisfactory here (so does this reviewer).

Several technical questions were raised, for which the authors gave detailed responses. In my opinion, the responses are satisfactory. So, the paper can be accepted, after a green signal from the ethics reviewer.

I could not find this reference anywhere on the Web: Yousheng Shu, Wangyu Hu, and Bingqian Chen. Selected Advanced Physics Problems, Volume II. Science Press, Beijing, 2008. ISBN 978703019356

**This paper is conditionally accepted provided the authors do the following for camera-ready**:
the authors need to provide evidence that this does not violate copyright and provide documentation suggesting substantial differences between their questions and source material.
They must also acknowledge the legal risk, if any, from using and releasing this data.

**Reviewer Concerns:**

Statistical significance: answered with multiple runs of each method

Use of PhysLib as a tool – this was not done. It would have been nice to establish baseline here, but in my judgment there is enough in the paper that it can be accepted without it.

Missing important related work, for which authors provide satisfactory comparison.

Various clarification questions.

One potentially hallucinated reference.

**Reviewer Scores:**

The 2 increased to 6, 8 remained 8. So, I do not need to speculate there. The 6 would have mostly stayed at 6 or moved to 7, since they did not ask any questions whose answers would have substantially changed their judgment.

---

### Decision · Program_Chairs · 2026-01-26

**Decision:**

Accept (Poster)

**Comment:**

Conditions for acceptance have been satisfied.